

# Assessment of healing dynamics in dental extraction sockets among non-diabetic, prediabetic, and type 2 diabetic patients: a comparative clinical investigation

Mohammed Saad Alqarni

Department of Oral & Maxillofacial Surgery and Diagnostic Sciences, College of Dentistry, Al Jouf University, Sakaka, Saudi Arabia

## ABSTRACT

**Background.** Diabetes mellitus is a chronic metabolic disorder characterized by persistent hyperglycaemia, affecting various metabolic processes and leading to multiple complications, particularly in wound healing. This study aims to evaluate the impact of diabetes on the healing of extraction sockets in non-diabetic, prediabetic, and type 2 diabetic patients in Saudi Arabia.

**Methodology.** A prospective observational study was conducted with 72 participants who were divided equally ($n = 24$ for each group) into three groups, *viz.* non-diabetic, prediabetic, and diabetic groups based on glycated hemoglobin (HbA1c) and random blood glucose levels. Tooth extractions were performed by an experienced maxillofacial surgeon. Healing outcomes were assessed by measuring extraction socket size, post-operative pain, discharge, swelling, infection, erythema, dry socket occurrence, and analgesic consumption over one week. Initially descriptive statistics were calculated and one way Analysis of Variance (ANOVA) test was done to compare the reduction in socket size between groups. The level of statistical significance was set at $P \le 0.05$.

**Results.** Out of 275 screened participants, 104 provided informed consent, and 72 completed the study. Significant differences were found in socket size reduction, with non-diabetic patients showing a 62.5% reduction, prediabetic 56.4%, and diabetic 48.6% ($p < 0.001$). Diabetic patients experienced more post-operative pain ($p = 0.039$) and a higher incidence of complications such as swelling, infection, and discharge, although not statistically significant ($p = 0.141$).

**Conclusion.** Diabetes significantly affects post-operative healing in dental extractions, leading to less socket size reduction, higher pain levels, and increased complications. These findings underscore the necessity for specialized post-operative care for diabetic patients, including stringent infection control and pain management strategies. Further research with larger sample sizes and extended follow-up periods is recommended to better understand the long-term impacts of diabetes on oral health.

Corresponding author
Mohammed Saad Alqarni,
dr.mohammed.alqarni@jodent.org

## INTRODUCTION

Diabetes mellitus (DM) is a chronic metabolic disorder characterized by persistent hyperglycemia due to defects in insulin secretion, insulin action, or both. This condition

leads to disturbances in carbohydrate, fat, and protein metabolism, resulting in various macrovascular and microvascular complications (*Schuster & Duvuuri, 2002*). According to the International Diabetes Federation (IDF), the global prevalence of diabetes in adults aged 20–79 years reached 10.5% (536 million individuals) in 2021, and this figure is expected to rise to 783 million by 2045. In Saudi Arabia, the prevalence of diabetes among adults was estimated to be 18.3% in 2021, making it one of the top 10 countries with the highest diabetes burden. This growing prevalence underscores the importance of understanding the oral health implications of diabetes (*International Diabetes Federation, 2021*).

Among the complications associated with diabetes, alterations in microvascular circulation are particularly concerning, as they can impair the body's inflammatory response, reduce leukocyte migration, and compromise tissue perfusion (*Lu et al., 2023*). These factors contribute to delayed wound healing and a higher risk of postoperative infections, particularly in dental and oral surgical contexts.

Dental extractions in diabetic patients pose unique challenges due to the impaired healing processes associated with the disease. Hyperglycaemia affects various stages of wound healing, including inflammation, proliferation, and remodelling (*Yang et al., 2022*). The compromised microvascular circulation in diabetic patients results in poor tissue perfusion, limiting the delivery of essential nutrients and immune cells to the extraction site. This can lead to prolonged inflammatory phases, slower re-epithelialization, and impaired tissue remodelling, ultimately delaying wound healing (*Ko, Sculean & Graves, 2021*). Existing literature on dental extractions in diabetic patients indicates an increased risk of complications such as delayed healing and postoperative infections (*Yang et al., 2022*; *Ko, Sculean & Graves, 2021*; *Horton & Barrett, 2021*; *Pezhman, Tahrani & Chimen, 2021*). Recent animal studies have demonstrated that even insulin-resistant (prediabetic) states delay gingival wound healing, which can be partially ameliorated with metformin (*Kominato et al., 2022*). Clinical studies in diabetic populations also show that poorer glycemic control is associated with delayed socket healing (*Murtaza Dad et al., 2025*). However, there is a noted lack of comprehensive studies specifically addressing these outcomes, especially in the Saudi population.

Additionally, diabetic individuals experience impaired immune responses due to hyperglycemia, which compromises leukocyte function and contributes to a chronic inflammatory state. This dysregulation increases the risk of infection and delays wound healing (*Horton & Barrett, 2021*; *Pezhman, Tahrani & Chimen, 2021*). These factors necessitate stringent infection control measures and close post-operative monitoring for diabetic patients undergoing dental extractions. While the effects of diabetes on wound healing are well established, prediabetes—a transitional state with impaired glucose homeostasis—remains underexplored in oral healing research. This gap highlights the need to evaluate post-extraction outcomes across the full glycemic spectrum.

The current study aims to address these gaps by evaluating the effect of prediabetes and diabetes on the healing of extraction sockets. By comparing non-diabetic, prediabetic, and type 2 diabetic patients, the study seeks to provide a comprehensive understanding of how varying levels of glycaemic control influence healing outcomes. The parameters assessed

include extraction socket size, postoperative pain, discharge, swelling, infection, erythema, and dry socket occurrence over one week.

This research will contribute valuable insights to the existing literature, particularly in the context of the Saudi Arabian population, where diabetes prevalence is rapidly increasing. Understanding these differences will aid in developing targeted strategies for managing diabetic patients undergoing dental extractions, ultimately improving clinical outcomes and patient care.

The null hypothesis of this study is that there would be no significant difference in socket healing parameters among non-diabetic, prediabetic, and diabetic individuals.

## MATERIALS AND METHODS

### Study design and setting

A prospective observational study was conducted in the Department of Oral and Maxillofacial Surgery and Diagnostic Sciences, Jouf University, KSA, over a one-year period from February 2024 to February 2025. Ethical approval was granted by the Local Committee of Bioethics, Jouf University on 28 January, 2024 (Ref: HAPO-13-S-001) and all the procedures in this study were in compliance with the Helsinki Declaration (9th version, 2013). At the time of treatment, patient's informed consent was obtained through a written consent form that included all the relevant details after elaborately explaining the treatment needs and goals along with the possible complications that could be expected.

### Sample size estimation

Sample size estimation was performed using GPower software (version 3.0). The calculation was based on an *F* test and ANOVA: Omnibus fixed effects one-way, considering three groups (non-diabetic, prediabetic, and diabetic) with equal sample sizes. An effect size of 0.39 was chosen, which was assessed from a study done by *Gadicherla et al. (2020)* which compared socket healing outcomes in diabetic and non-diabetic patients undergoing routine dental extractions. Their study reported healing differences based on glycemic status using clinical and radiographic parameters. To achieve a power of 80% and an alpha level of 0.05, the required sample size was estimated to be 66 participants. To account for a potential 10% attrition rate, the final sample size was increased to 72 participants, divided equally among the three groups ($n = 24$ for each group).

### Participants' selection

Participants aged 18 years and above, willing to participate in the study and undergoing single, non-surgical extractions were included. In cases where multiple extractions were performed, only the first extracted posterior (either premolars or molars) tooth socket was assessed.

Exclusion criteria encompassed individuals requiring trans-alveolar extraction, those needing extraction of anterior teeth (to reduce variability in socket morphology), those with conditions impairing wound healing (*e.g.*, HIV/AIDS (regardless of treatment status), chemotherapy, immunosuppressants, systemic steroids, chronic alcoholism, smoking, radiotherapy, bisphosphonates, and anticoagulants), benign or malignant jaw pathology,

patients on systemic antibiotic therapy at the time of enrolment and those unable to provide consent.

## Data collection and blinding

Pre-existing glycated hemoglobin (HbA1c) and random blood glucose levels were recorded to categorize participants into non-diabetic (HbA1c < 5.7%), prediabetic (5.7–6.4%), and diabetic (≥ 6.5% or confirmed diagnosis) groups based on American Diabetes Association criteria (*American Diabetes Association Professional Practice Committee, 2024*). These assessments are critical in determining the baseline metabolic status of the participants, influencing healing outcomes. All tooth extractions were non-surgical and performed using standard forceps technique by the same experienced surgeon under local anaesthesia, ensuring consistency across cases. Post-operative antibiotic use was not standardized but left to the discretion of the treating surgeon, reflecting real-world clinical practice. This pragmatic approach may introduce variability; however, it enhances the external validity of our findings.

Figure 1 shows selection process of the participants.

## Variables and assessment

On post-operative days 0 and 7, the extraction socket's largest diameter was measured using a calibrated probe by a single calibrated examiner. No duplicate measurements or inter-examiner validation was performed. Post-operative pain, discharge, swelling (graded as present if visible or palpable edema was noted in the adjacent soft tissue), infection, erythema (defined as the presence of visible redness around the socket margins, extending beyond the immediate extraction site), dry socket occurrence, and the number of analgesics consumed were recorded (*Centers for Disease Control and Prevention, 2023*). Pain was graded as 'no pain', 'mild pain', 'moderate pain' and 'severe pain. Criteria for dry socket (diagnosed based on severe pain at the extraction site, empty socket appearance with exposed bone, and absence of suppuration) and acute infection were defined according to *Aronovich et al. (2010)*.

## Statistical analysis

Data were analyzed using the Statistical Package for the Social Sciences version 20.0 (IBM Corp, Armonk, NY, USA). Data were entered into a MS Excel spreadsheet for editing and coding. Initially descriptive statistics were calculated and one-way ANOVA test was done to compare the reduction in socket size between groups. The level of statistical significance was set at $p \leq 0.05$.

# RESULTS

## Initial screening

A total of 275 participants were screened, out of which 120 met the inclusion criteria. This initial screening was crucial for ensuring that participants matched the study's requirements, such as age and the need for tooth extraction.

```
┌─────────────────────────────────────────────────┐
│   Patients assessed for eligibility (n = 275)    │
└─────────────────────────────────────────────────┘
                        ↓
┌─────────────────────────────────────────────────┐
│              Excluded (n = 203)                  │
│  -  Didn't meet inclusion criteria (n = 155)     │
│       -  Unwilling to participate (n = 39)       │
│            -  Other reasons (n = 9)              │
└─────────────────────────────────────────────────┘
                        ↓
┌─────────────────────────────────────────────────┐
│          Participants enrolled (n = 72)          │
└─────────────────────────────────────────────────┘
                        ↓
┌─────────────────────────────────────────────────┐
│               Group allocation                   │
│       -  Non-diabetic (n = 24)                   │
│         -  Prediabetic (n = 24)                  │
│           -  Diabetic (n = 24)                   │
└─────────────────────────────────────────────────┘
                        ↓
┌─────────────────────────────────────────────────┐
│                  Follow-up                       │
│  -  Completed by all the participants            │
│                (n = 72)                          │
│     -  No loss to follow-up (n = 0)              │
└─────────────────────────────────────────────────┘
```

**Figure 1  Patient flow in the study.**

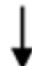

- *Enrolled participants:* 104 patients provided informed consent and were enrolled, ensuring ethical standards and participant willingness to partake in the study.

**Table 1  Baseline characteristics of the participants (age, gender, fasting glucose, HbA1c).**

| Group | N | Age (Mean ± SD, years) | Male n (%) | Female n (%) | Fasting glucose (mg/dL, Mean ± SD) | HbA1c (%, Mean ± SD) |
|---|---|---|---|---|---|---|
| Non-diabetic (Group 1) | 24 | 46.0 ± 11.2 | 14 (58.3%) | 10 (41.7%) | 90.5 ± 6.2 | 5.4 ± 0.2 |
| Prediabetic (Group 2) | 24 | 49.1 ± 10.3 | 11 (45.8%) | 13 (54.2%) | 110.8 ± 7.5 | 5.9 ± 0.2 |
| Diabetic (Group 3) | 24 | 47.5 ± 10.6 | 13 (54.2%) | 11 (45.8%) | 150.3 ± 18.6 | 8.1 ± 1.2 |
| **Total** | 72 | – | 38 (52.8%) | 34 (47.2%) | – | – |
| *P* values | – | 0.608, NS | 0.677, NS | | <0.001, S | <0.001, S |

| | | Age | | |
|---|---|---|---|---|
| **Group** | **N** | **Mean** | **Std. deviation** | *P* value |
| Group 1 | 24 | 46.0 | 11.2 | 0.608, NS |
| Group 2 | 24 | 49.1 | 10.3 | |
| Group 3 | 24 | 47.5 | 10.6 | |
| | | **Gender** | | |
| | | Male | Female | |
| Group 1 | N | 14 | 10 | |
| | % | 58.3% | 41.7% | 0.677, NS |
| Group 2 | N | 11 | 13 | |
| | % | 45.8% | 54.2% | |
| Group 3 | N | 13 | 11 | |
| | % | 54.2% | 45.8% | |
| **Total** | N | 38 | 34 | – |
| | % | 52.8% | 47.2% | – |

- *Final sample size:* The sample size was increased to 72 participants, divided equally into three groups (non-diabetic, prediabetic, and diabetic). This adjustment was made to ensure statistical power and account for potential dropouts.

*Baseline characteristics of participants:* The average age was comparable across groups ($p = 0.608$), and gender distribution did not significantly differ ($p = 0.677$). These demographics ensured that age and gender were not confounding factors in the analysis. In contrast, fasting glucose and HbA1c levels demonstrated highly significant differences between groups ($p < 0.001$) with the diabetic group exhibited the highest mean fasting glucose ($150.3 \pm 18.6$ mg/dL) and HbA1c levels ($8.1 \pm 1.2\%$), followed by the prediabetic group ($110.8 \pm 7.5$ mg/dL; $5.9 \pm 0.2\%$) and the non-diabetic group ($90.5 \pm 6.2$ mg/dL; $5.4 \pm 0.2\%$).

Table 1 shows the baseline characteristics of the participants.

## Socket size reduction

Intragroup comparison of mean socket size showed that among all the three study groups, the mean socket size at day 7 were significantly lower than that at day 0 ($p < 0.001$) (Table 2).

**Table 2  Intragroup comparison of mean socket size reduction.**

| Group | Day | Mean | N | Std. deviation | P value |
|---|---|---|---|---|---|
| Group 1 | Day 0 | 4.1 | 24 | 1.2 | <0.001, S |
|  | Day 7 | 1.5 | 24 | 0.6 |  |
| Group 2 | Day 0 | 3.9 | 24 | 1.3 | <0.001, S |
|  | Day 7 | 1.7 | 24 | 0.7 |  |
| Group 3 | Day 0 | 4.6 | 24 | 0.9 | <0.001, S |
|  | Day 7 | 2.4 | 24 | 0.7 |  |

**Table 3  Intergroup comparison of mean socket diameter in non-diabetic, pre-diabetic, and diabetic patients at day 0 and day 7.**

| Day | Group | N | Mean | Std. deviation | 95% Confidence interval for mean | | P value | Post hoc pairwise comparison |
|---|---|---|---|---|---|---|---|---|
|  |  |  |  |  | Lower bound | Upper bound |  |  |
| Day 0 | Group 1 | 24 | 4.1 | 1.2 | 3.6 | 4.6 | 0.141, NS | NA |
|  | Group 2 | 24 | 3.9 | 1.3 | 3.4 | 4.5 |  |  |
|  | Group 3 | 24 | 4.6 | 0.9 | 4.2 | 5.0 |  |  |
| Day 7 | Group 1 | 24 | 1.5 | 0.6 | 1.3 | 1.8 | <0.001, S | Gr1*Gr2–0.748, NS |
|  | Group 2 | 24 | 1.7 | 0.7 | 1.4 | 1.9 |  | Gr1*Gr3 - <0.001, S |
|  | Group 3 | 24 | 2.4 | 0.7 | 2.1 | 2.7 |  | Gr2*Gr3–0.002, S |

Intergroup comparison of mean socket size showed that the mean socket size at day 7 were significantly lower between groups ($p < 0.001$). The data showed a more reduction in socket size in group 1, followed by group 2 and group 3. The significant difference suggests that diabetes status may impact healing rates, with non-diabetic individuals showing greater reductions as shown in Table 3.

Intergroup comparison of mean percentage reduction in socket size from day 0 to day 7 was done using one way ANOVA test. A statistically significant difference was found among three study groups regarding mean percentage reduction in socket size ($p < 0.001$). *Post hoc* pairwise comparison using Tukey's test showed that mean percentage reduction in socket size among normal healthy group was found to be significantly more than that among pre-diabetics, which was further significantly higher than that among diabetics (Table 4).

### Post-operative pain

Intergroup comparison of different severity grades of pain showed that significantly a greater number of participants in group 3 reported to had severe type of pain, while significantly a greater number of participants in group 1 and group 2 reported to had mild type of pain ($p = 0.039$) as shown in Table 5. Thus, diabetic patients reported more pain, which may correlate with delayed healing and increased inflammation. The mean number of analgesic tablets consumed during the 7-day period was highest in the diabetic group

**Table 4  Intergroup comparison of mean percentage reduction in socket size from day 0 to day 7.**

| | N | Mean | Std. deviation | 95% Confidence interval for mean | |
|---|---|---|---|---|---|
| | | | | Lower bound | Upper bound |
| Group 1 | 24 | 62.5 | 8.4 | 58.9 | 66.1 |
| Group 2 | 24 | 56.4 | 13.6 | 50.7 | 62.2 |
| Group 3 | 24 | 48.6 | 10.7 | 44.1 | 53.1 |
| *P* value | | | <0.001, S | | |
| *Post hoc* pairwise comparisons | | | Gr 1 * Gr 2—0.149, NS | | |
| | | | Gr 1 * Gr 3—<0.001, S | | |
| | | | Gr 2 * Gr 3—0.045, S | | |

**Table 5  Intergroup comparison of different severity grades of pain.**

| Group | | Pain | | | | Total |
|---|---|---|---|---|---|---|
| | | No pain | Mild | Moderate | Severe | |
| Group 1 | n | 7 | 7 | 6 | 4 | 24 |
| | % | 29.1% | 29.1% | 25.0% | 16.7% | 100.0% |
| Group 2 | n | 7 | 10 | 6 | 1 | 24 |
| | % | 29.1% | 41.7% | 25.0% | 4.2% | 100.0% |
| Group 3 | n | 4 | 3 | 10 | 7 | 24 |
| | % | 16.7% | 12.5% | 41.7% | 29.2% | 100.0% |
| Total | n | 18 | 20 | 22 | 12 | 72 |
| | % | 25.0% | 27.8% | 30.6% | 16.7% | 100.0% |
| *P* value | | | 0.039, S | | | |

$(6.4 \pm 1.3)$, followed by the prediabetic group $(4.9 \pm 1.0)$, and lowest in the non-diabetic group $(3.7 \pm 0.9)$. The difference between groups was statistically significant ($p < 0.05$).

## Complications

The study monitored complications such as swelling, infection, discharge erythema and dry socket. Although there was no significant difference ($p = 0.141$), the incidence of complications was higher in diabetic patients, indicating potential risk factors associated with diabetes (Table 6).

## Summary of key findings

The study indicates that diabetic status affects post-operative healing, with diabetic patients showing less reduction in socket size and experiencing more complications and pain. These findings underscore the importance of tailored post-operative care for diabetic patients to mitigate risks and improve outcomes.

## DISCUSSION

This study compared early socket healing outcomes among non-diabetic, prediabetic, and diabetic patients. Unlike previous studies, such as *Gadicherla et al. (2020)* which compared

**Table 6  Occurrence of post-operative complications in non-diabetic, pre-diabetic, and diabetic patients.**

| Group | | None | Swelling | Infection | Discharge | Erythema | Dry socket | Total |
|---|---|---|---|---|---|---|---|---|
| Gr 1 | n | 20 | 3 | 0 | 0 | 1 | 0 | 24 |
| | % | 83.3% | 12.5% | 0.0% | 0.0% | 4.2% | 0.0% | 100.0% |
| Gr 2 | n | 18 | 1 | 1 | 0 | 2 | 2 | 24 |
| | % | 75.0% | 4.2% | 4.2% | 0.0% | 8.4% | 8.4% | 100.0% |
| Gr 3 | n | 12 | 2 | 1 | 1 | 4 | 4 | 24 |
| | % | 50.0% | 8.4% | 4.2% | 4.2% | 16.6% | 16.6% | 100.0% |
| Total | n | 50 | 6 | 2 | 1 | 7 | 6 | 72 |
| | % | 69.4% | 8.3% | 2.8% | 1.4% | 9.8% | 8.3% | 100.0% |
| *P* value | | | | 0.141, NS | | | | |

diabetic and non-diabetic patients, our study introduces prediabetic individuals as a distinct group. Moreover, the Saudi population adds further relevance given the high national prevalence of diabetes and prediabetes, underscoring the clinical importance of this tripartite comparison.

The key findings demonstrate that diabetic patients exhibited significantly slower socket size reduction and reported higher levels of post-operative pain compared to the other groups. While complication rates such as dry socket and erythema were not statistically different among groups, diabetic patients showed a trend toward greater inflammatory response.

The findings of this study align with the existing literature that underscores the impact of diabetes on post-operative healing, particularly in the context of dental extractions (*Ko, Sculean & Graves, 2021*; *Spampinato et al., 2020*; *Frisch et al., 2010*; *Thourani et al., 1999*; *Yamano, Kuo & Sukotjo, 2013*; *Nagy et al., 2001*). The delayed healing observed in diabetic patients, as evidenced by reduced socket size reduction and increased incidence of complications, echoes the results reported in various studies focused on diabetic wound healing in general and specifically in oral surgical contexts (*Brizeno et al., 2016*; *Lan et al., 2008*; *Dasari et al., 2021*; *Zhou et al., 2019*).

The significant reduction in socket size among non-diabetic patients (62.5%) compared to diabetic patients (48.6%) highlights the impaired healing capacity in diabetic individuals. This result is consistent with the findings of *Gadicherla et al. (2020)*, who reported similar trends in socket size reduction among different glycemic status groups. The observed differences may be attributed to impaired angiogenesis, reduced collagen synthesis, and prolonged inflammation, all of which are well-documented in diabetic patients.

Diabetic individuals exhibit impaired healing due to a combination of metabolic and vascular dysfunctions. Chronic hyperglycemia leads to the formation of advanced glycation end products, which interfere with collagen cross-linking and extracellular matrix stability. Additionally, diabetes impairs angiogenesis by reducing the bioavailability of nitric oxide and vascular endothelial growth factor, limiting oxygen and nutrient

supply to the healing site. Neutrophil chemotaxis, macrophage function, and fibroblast proliferation are also compromised, resulting in delayed granulation tissue formation and prolonged inflammation. Collectively, these disruptions lead to slower tissue regeneration and increased susceptibility to postoperative complications (*Brem & Tomic-Canic, 2007*).

Although the current study did not find statistically significant differences in post-operative complications such as swelling, infection, discharge, erythema and dry socket, the higher incidence in diabetic patients suggests a trend consistent with other studies. For instance, *Dallaserra et al. (2020)* reported an increased rate of postoperative infections and complications in diabetic patients undergoing dental procedures. The higher susceptibility to infections in diabetic patients is often linked to impaired immune responses and reduced leukocyte function, which are common complications of diabetes.

The study's findings on pain levels, with diabetic patients reporting more pain, corroborate other research indicating that diabetes exacerbates inflammatory responses. This increased pain perception in diabetic patients may be related to neuropathic changes and prolonged inflammatory mediators, as suggested by the work of *Schreiber et al. (2015)* who studied pain and inflammation in diabetic patients' post-extraction.

Our findings align with recent works showing delayed soft tissue healing in prediabetes and suggest that prediabetics represent an intermediate risk group (*Udeabor et al., 2023*). Interventions such as metformin treatment or adjunct therapies (*e.g.*, hyaluronic acid) have shown promise in improving healing outcomes in diabetic and prediabetic states (*Anari et al., 2019*). Clinically, these findings highlight prediabetes as a relevant risk factor that warrants early recognition and closer monitoring during dental extractions (*Muller et al., 2005*; *Lalla & Papapanou, 2011*).

The results underscore the need for clinicians to adopt tailored post-operative care strategies for diabetic patients to mitigate the risks of delayed healing and complications. This might include more rigorous monitoring, enhanced infection control measures, and potentially adjusting pain management protocols. Additionally, the study emphasizes the importance of patient education on the potential risks associated with dental procedures in the presence of diabetes, which could lead to better preoperative glycaemic control and, consequently, improved healing outcomes.

As the global prevalence of diabetes and prediabetes continues to rise, particularly in developing countries, these findings can inform dental practitioners and policymakers about the need for glycemic assessment, post-operative monitoring, and personalized care protocols for diabetic patients worldwide. Future multicenter studies involving diverse populations are warranted to validate and generalize these results across different healthcare systems and genetic backgrounds.

This study highlights the importance of interdisciplinary collaboration in managing diabetic patients. Oral surgeons, endocrinologists, and primary care providers must work closely to optimize the patient's glycaemic control before, during, and after surgery. This integrated approach not only improves oral health outcomes but also contributes to better overall health and quality of life for diabetic patients.

### Limitations

This study has several limitations that should be acknowledged. First, the follow-up duration was limited to 7 days, preventing assessment of complete bone healing. Second, the sample size was calculated to detect moderate effects but may be underpowered for smaller differences or multivariate analyses. Pain was assessed using a categorical scale for simplicity and feasibility in a busy outpatient clinical setting. While this method allowed reliable patient reporting, we acknowledge that validated tools such as the visual analogue scale could provide greater precision and recommend their use in future studies. Thirdly, outcome measurements were performed by a single, unblinded examiner without duplicate validation, which introduces potential performance bias and prevents error correction. Additionally, the study did not stratify data based on tooth type (*e.g.*, molar *versus* premolar), which can influence socket size and healing. Concomitant medication uses and other clinical variables were not controlled for, which may act as confounders. The statistical analysis employed one-way ANOVA, which does not account for within-subject changes over time; a mixed-model ANOVA would have been more appropriate to account for within-subject changes over time, the limited sample size and study design did not allow for this advanced model. Future studies should use multicenter studies with larger samples, validated measurement tools, blinded assessors, and extended follow-up with more rigorous statistical models to validate and generalize these results.

## CONCLUSION

Diabetic patients exhibited slower socket size reduction and reported higher post-operative pain compared to non-diabetic and prediabetic individuals. These findings suggest that glycemic status plays a critical role in early post-extraction healing and highlight the need for tailored perioperative care in diabetic and prediabetic patients. Clinically, our results emphasize the need for tailored follow-up protocols for both diabetic and prediabetic patients. Closer post-extraction monitoring, reinforced infection-control strategies, and individualized use of antibiotics and analgesics may help mitigate delayed healing and complications in these at-risk groups. These measures could mitigate delayed healing and reduce the risk of complications in patients with impaired glucose metabolism.

### Funding

The author received no funding for this work.

### Competing Interests

The author declares that they have no competing interests.

### Author Contributions

- Mohammed Saad Alqarni conceived and designed the experiments, performed the experiments, analyzed the data, prepared figures and/or tables, authored or reviewed drafts of the article, and approved the final draft.

## Human Ethics

The following information was supplied relating to ethical approvals (*i.e.*, approving body and any reference numbers):

The Local Committee of Bioethics, Jouf university, KSA, provided approval to carry out this project (7450) with approval number HAPO-13-S-001.

## Data Availability

The data is available in the Supplemental Files.

## Supplemental Information

Supplemental information for this article can be found online at http://dx.doi.org/10.7717/peerj.20361#supplemental-information.

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
