# Peer review of "Assessment of healing dynamics in dental extraction sockets among non-diabetic, prediabetic, and type 2 diabetic patients: a comparative clinical investigation"

_PeerJ, doi:10.7717/peerj.20361_

## Round 0.1 · original submission · Major Revisions

· Academic Editor

Major Revisions

Dear authors,

Your manuscript has been reviewed by five peer reviewers. Please ensure that you address all of their comments and questions thoroughly to help avoid multiple rounds of revision.

Thank you for considering PeerJ for the publication of your work.

**Language Note:** The review process has identified that the English language must be improved. PeerJ can provide language editing services - please contact us at [email protected] for pricing (be sure to provide your manuscript number and title). Alternatively, you should make your own arrangements to improve the language quality and provide details in your response letter. – PeerJ Staff

Reviewer 1 ·

Basic reporting

The English is clear and unambiguous. References, background information, and context are sufficient and related to the study's topic. The structure of the article and tables is appropriate. The raw data were shared. The presentation is self-contained, with relevant results to test the hypotheses.

Experimental design

The research is original and falls within the journal's objectives and scope. The research question is well-defined, relevant, and significant. The research fills an identified knowledge gap. The research is rigorous and was conducted with high technical and ethical standards. The methods have been described in sufficient detail and information to allow replication of the study.

Validity of the findings

It has a significant impact and novelty. All underlying data have been provided; they are robust, statistically sound, and controlled. The conclusions are well presented, linked to the original research question, and limited to supporting results.

Reviewer 2 ·

Basic reporting

The manuscript was written clearly. More literature references to be cited in the text. Raw data is shared.

Experimental design

Experimental design is fine.

Validity of the findings

The results were elaborated. Discussion should explain the impact of this study on the global population, as this study is limited only to Saudi Arabia.

Additional comments

1. The authors should justify the rationale for categorizing participants into non-diabetic, prediabetic, and diabetic groups based on HbA1c and random blood glucose levels.
2. How were the tooth extractions performed, and what factors were controlled for to ensure consistency?
3. Why was only one week of post-operative monitoring chosen? Could a longer follow-up period have provided more insights?. The authors should clarify this
4. Why was the percentage of socket size decrease lower in diabetic individuals than in non-diabetic and prediabetic patients?
5. What could be the clinical implications of the findings regarding post-operative pain in diabetic patients?
6. How did the authors determine the sample size, and does this sample size have enough power to detect smaller differences?
7. How could the findings of this study influence clinical guidelines for dental surgeons when treating diabetic patients?
8. What were the potential limitations of this study
9. Ethical clearance year and date to be mentioned

Reviewer 3 ·

Basic reporting

Using an effect size to 7 decimal points is not necessary.

Table 4 is unlikely to be documenting socket size, as some means are 1.5 mm. More likely that they document socket reduction. Some comment is needed on the accuracy of your measurements (were other experimenters involved to provide other duplicate readings)?

Experimental design

Did you exclude those undergoing surgical extractions? Did anyone receive multiple extractions, and if so, which socket was measured? Socket reduction may vary depending on which tooth is to be extracted (molar sockets are larger than incisor sockets). Your analysis does not take this into account.

Similarly, you do not mention if you screened your patients for pre-extraction or post-extraction antibiotics. The HbA1c data was not included in the analysis, but if regression analysis was used, it might have been a significant factor.

Validity of the findings

As well as the above criticisms, you do not state that patients with medical conditions likely to affect would healing (eg, anti-cancer therapy) were excluded.

Additional comments

"Blinded" measurements (ie, where the assessor does not know the diabetic status of the patients) would have increased confidence in the data.

Reviewer 4 ·

Basic reporting

The authors reported the socket healing and complications of patients with different diabetic statuses. The structure of the article does not match the standards, with signs of AI writing style. The tables are too many tables. Results are underreported, particularly on glucose homeostasis, which is lacking.

Experimental design

Methods: Sample size – effect size was derived from Gadicherla et al. The phrase “among the group” is vague. Can the authors define what kind of subjects were involved in Gadicherla et al.’s study?
Exclusion criteria: HIV/AIDS – untreated or treated?

Blinding: The methods of blinding were not sufficiently described. Was the dentist examining the patients aware of their glucose homeostasis status?

Statistical analysis: I do not agree with this paper's statistical analysis strategy. The study design adopted a time x group design but used one-way ANOVA to analyse the data. A mixed-design ANOVA would be more suitable to determine the between- and within-subject effects.

Validity of the findings

I do not find the results credible as the statistical approach is incorrect, especially time difference was analysed using one-way ANOVA.

Additional comments

Abstract: Can you indicate the number of subgroups?

Introduction: Please indicate the years for a survey when citing epidemiological studies.
The study parameters should be defined in the methods section.

Methods: Sample size – effect size was derived from Gadicherla et al. The phrase “among the group” is vague. Can the authors define what kind of subjects were involved in Gadicherla et al.’s study?

Exclusion criteria: HIV/AIDS – untreated or treated?

Blinding: The methods of blinding were not sufficiently described. Was the dentist examining the patients aware of their glucose homeostasis status?

Statistical analysis: I do not agree with the statistical analysis strategy of this paper. The study design adopted a time x group design but used one-way ANOVA to analyse the data. The more suitable approach will be mixed-design ANOVA to determine the between and within-subject effects.

Results: I do not find the results credible as the statistical approach is incorrect.
Why were no parameters on glucose homeostasis presented?

Other minor points on results:
Prepare a flow chart summarizing the recruitment process, instead of Table 1.
There are too many tables. Please organize them and reduce in number.
The way of writing is problematic, as the paragraph begins with aspects of being examined: explanation. This is similar to text generated by AI.

Discussion: The first 4 paragraphs contain information already covered in the Introduction. Please condense them. Start with an overview of the results.

Reviewer 5 ·

Basic reporting

I suggest that the English language be improved throughout the manuscript, avoiding informal language and repeated words.
In the methods section, some references are necessary to validate and allow reproducibility of the results.

Experimental design

The introduction should be improved and restructured, adding important information to justify the study, making the objective clearer and more concise, and including the null hypothesis to be tested.
Methods do not provide sufficient details and information to replicate. Some modifications are necessary.
Modifications should also be made to the discussion and results sections to improve the manuscript.

Validity of the findings

Conclusions were not well formulated, so they should be rewritten in an objective and clear manner, based on the results.

Additional comments

Comments to the authors:

Abstract
- It is important to briefly mention the inclusion and exclusion criteria.
- Inform when and where exactly the study was conducted.
- Is it necessary to include how many patients are in each group?
- There is no information on how the pain assessment was performed and how the size of the alveolus was measured.
- A brief mention of limitations would reinforce scientific credibility.

Introduction
- There is excessive repetition of ideas, for example: Compromised microcirculation is mentioned twice, with similar explanations, and the effects of hyperglycemia on the immune response appear in two paragraphs with little new between them. I suggest consolidating these ideas into a single, well-structured paragraph with clear causal logic.
- Although relevant, the explanation about the phases of healing and inflammatory cytokines is too in-depth for an introduction.
- The paragraph about the immune system appears separate from the previous one, when it could be integrated into the explanation of the impact of DM on healing.
- I suggest that this section delve deeper into tooth extractions and local healing. - Including specific authors or systematic reviews would help to reinforce the gap in the literature.
- Rewrite the objective clearly.
- Some phrases are informal and should be replaced, “Perhaps more concerning…” “This can lead to…”
- “7 million diabetics” and “tenfold increase” are mentioned, but without indicating the study or year.
- The topics at the end (“Study Parameters”) are unnecessary in the introduction.
- The null hypothesis was not stated.

Methodology
- It is important to make it clear that the groups were formed based on pre-existing HbA1c levels, not randomized.
- How exactly was “alveolar size” measured?
- The authors mention: “no pain”, “mild”, “moderate”, “severe”, but do not cite a validated scale such as VAS (Visual Analog Scale) or NRS (Numeric Rating Scale). This compromises the external validity of the pain data. - It is not clear who performed the postoperative measurements (surgeon, blinded evaluator?)
- Criteria for infection, swelling, and erythema are not described in detail (except dry socket). Use standardized and referenced clinical criteria.
- Follow-up is only 7 days — it would be ideal to justify this or mention it as a limitation at the end of the section or in the discussion.
- There is no mention of the treatment of missing data or participant dropout.

Results
- The results are described in general terms, but the raw data, with mean, standard deviation, confidence interval, and p-values, are missing.
- The section could be better organized by internal subheadings, such as: Participant Flow and Demographics, Socket Size Reduction, Postoperative Pain, Complication Rates, and Summary of Key Findings.
- The number of analgesics consumed by the group was not presented in the results.
- Data on dry socket and erythema — cited in the method, missing here.
- If you have enough data, you could analyze whether age, sex, or continuous HbA1c impacted any outcome, even if in an exploratory way.

Discussion
- The authors begin by talking about clinical challenges in general. It is more important to start by reviewing the objective of the research to connect with the findings and justify whether the null hypothesis was accepted or rejected.
- There was no significant difference in complications, so the authors suggest a trend. Could this be due to the short follow-up time? Could there be underreporting of events? Was the sample sufficient to detect these secondary differences? The lack of significance does not rule out a relevant clinical effect.
- There is a brief mention of angiogenesis, collagen, and inflammation; I suggest expanding the explanation of why diabetics heal worse, based on physiology.
- It is important to explain whether pre-diabetics showed an intermediate pattern between healthy individuals and diabetics, and what this implies clinically.
- Was there any linear correlation between HbA1c and alveolar size on day 7?
- The authors suggest glycemic monitoring and control, but it could be more specific, such as: Prophylactic use of antibiotics in decompensated diabetics? Reinforced suture and hemostasis techniques? Tooth extractions under strict glycemic control (<7% HbA1c)? Assessment of the ideal time for postoperative return?

Conclusion
- Rewrite the conclusion in a clear and concise manner, highlighting the main findings and emphasizing the authors' opinion.

---

## Round 0.2 · Major Revisions

· Academic Editor

Major Revisions

Dear authors,

Please carefully consider and respond to the comments provided by the reviewers.

Reviewer 1 ·

Basic reporting

The English is clear and unambiguous. References, background information, and context are sufficient and related to the study's topic. The structure of the article and tables is appropriate. The raw data were shared. The presentation is self-contained, with relevant results to test the hypotheses.

Experimental design

The research is original and falls within the journal's objectives and scope. The research question is well-defined, relevant, and significant. The research fills an identified knowledge gap. The research is rigorous and was conducted with high technical and ethical standards. The methods have been described in sufficient detail and information to allow replication of the study.

Validity of the findings

It has a significant impact and novelty. All underlying data have been provided; they are robust, statistically sound, and controlled. The conclusions are well presented, linked to the original research question, and limited to supporting results.

Additional comments

All reviewers' comments were addressed.

Reviewer 2 ·

Basic reporting

No Comments as the manuscript has been reviewed by me

Experimental design

No Comments as the manuscript has been reviewed by me

Validity of the findings

No Comments as the manuscript has been reviewed by me

Additional comments

The authors have incorporated the suggestions provided by the reviewers

Reviewer 4 ·

Basic reporting

The writing of the manuscript and expression have signficantly improved.

Experimental design

Analysis: The refusal of the authors to amend the analysis based on mixed design ANOVA frustrated the reviewers. This means that I cannot trust the analysis due to inflated errors.

Validity of the findings

Limitations that should be acknowledged:
No blinding – potential performance bias.
No duplication measurements – Cannot correct errors of measurement.

Additional comments

Other comments:
In Table 1: Characteristics of the subjects, basic reporting of the glucose homeostasis parameters (fasting glucose, HbA1c levels should be reported).
In many of the tables, the decimal places of the data have not been corrected. For age, one decimal place would be sufficient (mean and SD). For socket size, it depends on the precision level of the tool used. The decimal places should not be more than precision level of the tool.

Reviewer 5 ·

Basic reporting

- The manuscript addresses a clinically relevant question and presents original data comparing socket healing in non-diabetic, prediabetic, and diabetic patients. The inclusion of prediabetic individuals is particularly valuable, as this group is often overlooked in the literature. However, some modifications are needed to strengthen this manuscript.
- I suggest the introduction be more concise, especially when mentioning the prevalence of diabetes, highlighting the research gap.
- I recommend that the authors more directly report how the manuscript differs from previous studies (e.g., Gadicherla et al., 2020) and why the Saudi population adds relevance.
- Some references are pertinent, but I suggest the authors include more recent literature on oral healing in diabetics and prediabetes as a risk factor, reinforcing the originality of the tripartite comparison.
- The tables and figures are clear, but Figure 1 (flowchart) could be improved with more details on sample losses in each phase.
- I recommend that the authors review the formatting of Tables 3 and 4, as some characters appear disfigured.

Experimental design

- The measurements were performed by a single, unblinded examiner, which may introduce bias. It would be interesting to acknowledge this limitation and for the authors to suggest strategies to overcome it in future studies.
- Why was postoperative antibiotic therapy not standardized? This may have influenced healing and complication outcomes. Therefore, the authors should better justify this decision and discuss the potential impact on the results.
- The categorical scale used for pain outcomes is simple. Why were more accurately validated instruments, such as the VAS, not used?
- The authors used only one-way ANOVA; however, considering repeated measures (days 0 and 7), more robust analyses such as repeated measures ANOVA or mixed models would be more appropriate. I recommend mentioning this and reanalyzing the data.
- The authors should acknowledge the lack of control for clinical variables, the lack of stratification by type of tooth extracted, and the lack of adjustment for concomitant medication use.

Validity of the findings

- The data presented support the main conclusion that diabetes impairs post-extraction healing. However, some aspects should be considered.
- The analyses of complications (Table 6) showed no statistical significance, but the conclusion suggests a tendency toward a higher risk in diabetics. I recommend that the authors rephrase this sentence to clarify that this is a non-significant trend.
- The results obtained in pre-diabetics require further discussion. I also suggest that the authors emphasize the clinical importance of this intermediate condition as a risk factor.
- The interpretation of the findings should be more balanced, highlighting the need for multicenter studies with larger samples and longer follow-up to confirm the robustness of the results.

Additional comments

- The authors should add and acknowledge in the limitations section some aspects related to selection bias, as many patients were excluded; the short follow-up period of 7 days, which makes it difficult to assess the complete bone healing process; and the lack of multivariate analysis to control potential confounders.
- Furthermore, it would be helpful if the authors included more detailed practical implications for dentists, such as specific follow-up protocols and guidelines on the use of antibiotics or analgesics in diabetic and pre-diabetic patients.

---

## Round 0.3 · accepted · Accept

· Academic Editor

Accept

The authors have thoroughly addressed all reviewer comments and have significantly strengthened the manuscript. I am pleased to accept the paper for publication in its current form.

Reviewer 5 ·

Basic reporting

The authors have thoroughly addressed all reviewer comments and strengthened the manuscript.

Experimental design

The authors have thoroughly addressed all reviewer comments and strengthened the manuscript.

Validity of the findings

The authors have thoroughly addressed all reviewer comments and strengthened the manuscript.

Additional comments

The authors have thoroughly addressed all reviewer comments and strengthened the manuscript.